# Spatial Effect of Digital Economy on Particulate Matter 2.5 in the Process of Smart Cities: Evidence from Prefecture-Level Cities in China

**DOI:** 10.3390/ijerph192114456

**Published:** 2022-11-04

**Authors:** Jingrong Tan, Lin Chen

**Affiliations:** School of Economics, Zhejiang University of Technology, Hangzhou 310023, China

**Keywords:** digital economy, particulate matter 2.5 emissions, spatial Durbin model, smart city pilot policy, spatial difference-in-differences model

## Abstract

During the COVID-19 pandemic, the digital economy has developed rapidly. The airborne nature of COVID-19 viruses has attracted worldwide attention. Therefore, it is of great significance to analyze the impact of the digital economy on particulate matter 2.5 (PM_2.5_) emissions. The research sample of this paper include 283 prefecture-level cities in China from 2011 to 2019 in China. Spatial Durbin model was adopted to explore the spatial spillover effect of digital economy on PM_2.5_ emissions. In addition, considering the impact of smart city pilot (SCP) policy, a spatial difference-in-differences (SDID) model was used to analyze policy effects. The estimation results indicated that (1) the development of the digital economy significantly reduces PM_2.5_ emissions. (2) The spatial spillover effect of the digital economy significantly reduces PM_2.5_ emissions in neighboring cities. (3) Smart city construction increases PM_2.5_ emissions in neighboring cities. (4) The reduction effect of the digital economy on PM_2.5_ is more pronounced in the sample of eastern cities and urban agglomerations.

## 1. Introduction

The wide spread of COVID-19 in early 2020 has changed the way we live and work. In an environment where the epidemic was spreading, new digital business models such as online office, online shopping, online teaching, and online video were emerging, and the digital economy have been developing rapidly [1]. In 2016, the G20 Digital Economic Development and Cooperation Initiative clarified the concept of the digital economy, highlighting digital information and knowledge and their role as carriers of information networks. It also identified new information and communication technologies (ICT) as a critical driver for upgrading the economy’s structure [2]. In 2020, the value of China’s digital economy reached CNY 39.2 trillion, accounting for 7.8% of the gross domestic product (Data from a white paper on the development of China’s digital economy published by the China Academy of Information and Communication Technology). During the global epidemic outbreak crisis, the development of the digital economy has provided a strong impetus and guarantee for economic development.

Medical studies have indicated that the main channels of transmission of COVID-19 viruses include contact transmissions, mother-to-child transmission, and aerosol transmission [3]. Aerosol transmission refers to the mixing of droplets in the air to form an aerosol which, when inhaled, causes infection. Delicate particulate matters are airborne particles equal to or less than 2.5 microns in diameter, which can be suspended in the air for a more extended period and have a more significant impact on human health and the quality of the atmospheric environment. During the period of COVID-19 outbreak, exposure to PM_2.5_ increased the risk of more severe symptoms at the time of infection [4], including death (Harvard T.H. Chan School of Public Health. (n.d.). Coronavirus and air pollution. https://www.hsph.harvard.edu/c-change/subtopics/coronavirus-and-pollution/, accessed on 25 August 2022). According to the air quality report 2021 (https://www.iqair.cn/cn/world-air-quality-report (accessed on 25 August 2022)), 143 of 1347 cities in East Asia, had a PM_2.5_ greater than seven times the World Health Organization (WHO) air quality objective value, and all 143 of these cities are from China, where average PM_2.5_ concentration in 2021 was 6.5 times the WHO annual air quality objective value. In 2021, China ranked as the 22nd dirtiest country in the world. A series of facts show that air pollution caused by PM_2.5_ emissions is a significant problem in China and, as such, poses a considerable threat to public health and sustainable economic development. Especially in the case of widespread COVID-19 virus transmission, air quality has a direct impact on the rate of virus transmission [4].

Since the deepening of reform and opening up in China, the rapid economic development urbanization level has been increasing. The urbanization process over the past 40 years had shown that the problem of high PM_2.5_ concentrations was mainly concentrated in cities [5,6]. Unfortunately, China’s brutal model of economic development over the past decades has caused irreversible environmental and energy crises [7]. Based on the transmission characteristics of the new coronavirus and the development of the digital economy in the post-epidemic era, it is essential to analyze the impact of the digital economy on urban PM_2.5_ emissions. In addition, in August 2013, China’s Ministry of Industry and Information Technology implemented a pilot policy on smart cities as an innovation to the traditional urban governance model. There is a quasi-natural experimental scenario for empirical analysis of whether innovations in urban governance models in the development of the digital economy can reduce PM_2.5_ emissions.

Employing panel data of China’s 283 cities from 2011 to 2019, this paper empirically explores the effects of the digital economic development and SCP policy on PM_2.5_, adopting the spatial Durbin model and the spatial difference-in-difference (SDID) model, respectively. We mainly analyze the following question: Does digital economy development reduce PM_2.5_ emissions? Does the digital economy have spatial spillover effect on PM_2.5_? Have SCP policy reduced PM_2.5_ emissions? Does smart city construction have a spatial spillover effect on PM_2.5_ emissions?

The main contributions of this study can be summarized as follows: (1) Existing studies on the digital economy have not standardized the definition and measurement of the digital economy. This study employs an indicator system to construct digital economy indicators at the city level in China through the entropy value method to complement the digital economy research. (2) This paper quantified the impact of smart city construction on PM_2.5_ emissions in a quasi-natural experiment using China’s SCP policy. Thus, it expands the existing research perspective on smart cities and adds to the existing literature by linking environmental pollution to smart city construction. (3) The spatial spillover effects of digital economy and smart city construction are vital issues that may be neglected in existing studies. This study also analyzes the impact of the digital economy and smart city construction on PM_2.5_ emissions in pilot cities and neighboring non-pilot cities. 

The structure of this paper is arranged as follows. Section 2 is a literature review. Section 3 shows the research design and empirical analysis model. Section 4 further explains the spatial characteristics of the digital economy and PM_2.5_ emissions. Section 5 presents and discusses the empirical results. Section 6 carries out an analysis of spatial heterogeneity. Section 7 summarizes the conclusion.

## 2. Literature Review

Reviewing the digital economy, Tapscott (1996) indicated that the digital economy explains the relationship between the new economy, new business, and new technology [8]. Rouse (2016) defined the digital economy as the economy based on digital technology and considered the digital economy as a global network of economic activities supported by information and communication technologies [9]. Further, Dahlman et al. (2016) explained that the digital economy integrates a variety of generic technologies, a range of economic and social activities carried out through the internet, including digital technologies based on physical infrastructure, used to access devices and their applications to provide conditions [10]. 

With the rapid development of the internet, big data, cloud computing, blockchain, and other modern information technologies, the digital economy was increasingly used in life and production [11]. Especially in the light of COVID-19, the digital economy was promoted more widely. The digital economy has become a significant driver for the development of countries in the future [12,13]. As a result, the digital economy has attracted widespread attention from political and academic circles. 

First, some of the studies point to the digital economy as a driver for the development of technology promotion, which improves economic development, social productivity, and resource allocation throughout society [14,15,16]. On the one hand, the emergence of the internet has reshaped operational and organizational structures, deflating spatial and temporal constraints and enabling greater automation of traditional manufacturing processed, leading to lower production costs, and higher productivity [14,17,18]. On the other hand, the digital economy promotes connectivity, mere sharing and innovative collaboration between economic agents, thus promoting digital technology empowerment [19,20,21]. With technological innovation as the driving force of economic action, the digital economy faces a broader scope for development against the backdrop of the ongoing advancement of global sustainable development goals. 

Second, with the advent of the internet, the digital economy has had a significant impact on changes in the way societies live. For example, online shopping has eliminated geographical restrictions, allowing people worldwide to buy the same products and services, and promoting a global balance between supply and demand [12]. In addition, the digital economy has played an irreplaceable role in the context of the spreading of COVID-19. Online meetings, online offices, and online teaching provide a condition for isolating groups [22,23]. The digital economy will play an increasingly important role in future development during a century of unprecedented change.

Third, the digital economy development also influenced the environment. With global environmental constraints, increasing applications of digital technologies are emerging in the energy and environmental sectors [16,18]. The information integration capabilities of digital technologies help energy companies to improve the efficiency, to reduce production costs, and to extend the benefits of clean energy [24]. In addition, scholars in the field have shifted from concentrating on whether the digital economy is good for the environment to focus on how the digital economy is good for the environment.

### 2.1. Research on Digital Economy and Emissions Nexus

To meet the requirements of adapting to green development and addressing global environmental issues, the digital economy has become a significant factor in promoting quality economic development [13,18]. The main aspects are summarized below.

The development of digital economy contributes to the technological improvement of the energy industry. Litvinenko (2020) uses the example of the mineral industry in the Russian Federation for his analysis and illustrates that digital systems reduce production costs and improve the quality of human capital to achieve organizational efficiency gains [25]. In addition, using Japan as an example for analysis, Ahl et al. (2020) suggested that in the energy sector, blockchain can reduce transaction costs, facilitate distributed, peer-to-peer transactions, and create an innovative ecosystem for energy transformation [26]. Park and Heo (2020) indicated that the energy data sharing mechanism and efficient regulatory system established by ICT has facilitated the rapid development of the energy sector in power industry in Korea [27]. One of the essential ways to reduce emissions is to improve energy efficiency [7,18,28,29].

### 2.2. Smart City Construction and Emissions Nexus

As the foundation of urban digital economy development, the construction of a smart city takes an exogenous influence, and it is of great significance to investigate its influence on emissions [13,16].

Smart cities are considered to be a transformation of city building and management, with the integration of urban resources and information technology at their core [30]. Then, through the integration of urban resources and information technology, smart city construction effectively improves air quality and has a significant positive spillover effect on air pollution in neighboring cities, such that most of it can be attributed to the technological impact [31]. On one hand, information technology can break through regional restrictions and facilitate spreading technology and knowledge across regions. This enables innovation dividends to be shared, which drives the production and lifestyle of other cities around the smart city in the direction of intelligence [27,32]. On the other hand, the construction of smart cities promotes the upgrading of urban industries structure and frees up space for the development of high-tech industries by shifting energy-intensive and pollution-intensive industry within the region [16,33]. 

A review of the literature revealed that there are still two gaps in the research on this subject. First, even if the research on the digital economy have been around for a long time, there is still a study gap in the definition and measures standard. Second, although existing studies have analyzed impacts of digital economy on the environment, there is still a research gap in the effect of digital economy and smart city construction on PM_2.5_ emissions. The following study will fill the above gaps.

## 3. Research Design

### 3.1. Methodology

#### 3.1.1. Spatial Autocorrelation Model

Spatial autocorrelation refers to the potential interdependence between the observation variables within the same distribution region. Especially, areas with similar locations have similar variable values. If high values are adjacent to high values, or common values are adjacent to common values, positive spatial autocorrelation exists. If high values are adjacent to low values, there is a negative spatial correlation. If high values and common values are published randomly, there is no spatial correlation [34]. The global Moran’s I index was used to conduct spatial autocorrelation analysis of urban PM_2.5_ emissions, and the relevant formula as follows [35].
(1)$$I=\frac{n\sum_{i=1}^{n}\sum_{j=1}^{n}W_{ij}\left|c_{i}-\bar{c}\right|\left|c_{j}-\bar{c}\right|}{\sum_{i=1}^{n}\sum_{j=1}^{n}W_{ij}\sum_{i=1}^{n}\left|c_{j}-\bar{c}\right|}$$

The value of *I* ranges from −1 to 1. A positive value of *I* indicates a positive autocorrelation in the neighboring space. The larger the *I*, the stronger the spatial correlation. Conversely, a negative value of *I* represents a negative autocorrelation in the adjacent area. If *I* is equal to 0, it indicates no spatial autocorrelation in urban carbon emissions [36]. *W_ij_* denoted the neighborhood weight matrix, and *n* represented the total number of cities. *c_i_*, *c_j_* represented the carbon emissions of city *i* and city *j*, respectively. $\bar{c}$ defined the average value of carbon emissions.

#### 3.1.2. Spatial Markov Chains

A Markov chain is mainly used to analyze the continuous attribute values of an index in different periods. Usually, the data level division is used to estimate the probability distribution and change of each type, and the evolution and development process of geographical phenomena are approximated as Markov processes [37]. A particular kind of distribution at time *t* is represented by the state probability vector of $E_{t}=[E_{1,t},E_{2,t},\cdots ,E_{k,t}]$ of 1 *× k*, and the whole state transition process is described by the probability value *k × k*, as the Markov probability transition matrix *M_ij_* [38,39]. *M_ij_* represented the probability that a spatial unit of type *i* at time *t* becomes of type *j* at time *t* + 1.
(2)$$M_{ij}=n_{ij}/n_{i}$$
where *n_ij_* represents the number of type *i* at time *t* that become type *j* at time *t* + 1, and *n_i_* represents the sum of all kinds of *i* during the study period.

The regional correlation and dependence of the digital economy in geographic space could not be ignored. The spatial Markov chain combined with the concept of spatial lag made up for the lack of missing spatial interaction in the static Markov chain. The spatial Markov chain introduce the spatial weight matrix to calculate the weighted average attributes of adjacent regions to analyze the neighborhood conditions of spatial nits. If the Markov chain has *N* possible states, the size of the transition matrix is of order *N*. To analyze the state transition trend and dynamic evolution characteristics of the research object, the digital economy level firstly was divided into *N* types by the method of the natural break point. Secondly, the corresponding Markov probability transition matrix is constructed to reflect the dynamic characteristics of the development of the digital economy. Considering the mutual influence of the development of the digital economy in the neighborhood, space factors are considered in the Markov transition matrix [40], as shown in the following model:(3)$$value=\sum Digital_{i}W_{ij}$$
where, the *value* represents the spatial lag value, measuring the development level of the digital economy in the spatial neighborhood, and obtained by the natural break point method. *Digital_i_* represents the digital economy level in city *i*. *W_ij_* denoted the spatial weight matrix.

#### 3.1.3. Spatial Econometric Model

In this study, the spatial Durbin model (SDM) is used to introduces the spatial lag term of dependent variables and the spatial lag term of the independent variables as independent variables. The following model analyzes the relationship between the digital economy and PM_2.5_ emissions.
(4)$$\ln pm_{it}=\alpha_{0}+\beta \sum_{j=1}^{n}W_{ij}\ln pm_{it}+\gamma \ln(X_{it})+\lambda \sum_{j=1}^{n}W_{ij}\ln X_{it}+\mu_{i}+\upsilon_{t}+\varepsilon_{it}$$
where, *pm_it_* is PM_2.5_ emissions per unit of output in city *i*, and *β* is the spatial lag regression coefficient, indicating the degree of mutual influence of spatial neighborhood PM_2.5_ emissions. *X_it_* includes total independent variables in city *i*. *γ* is the regression coefficients of independent variables, and *λ* spatial lag regression coefficients of independent variables. *α*_0_, *μ_i_*, and *ν_t_* represent constant term, individual fixed, and time fixed effects, respectively. *ε_i__t_* is the random disturbance term. *W_ij_* is the spatial weight matrix. In this study, three spatial weights are used to analyze the spatial Durbin model. The first matrix ($W_{ij}^{1}$) is the spatial adjacency matrix, which takes the value of 1 for neighboring cities and 0 otherwise. The second matrix ($W_{ij}^{2}$) is the economic distance matrix, constructed based on the inverse of the gap between the per capita GDP of two cities. The third matrix ($W_{ij}^{3}$) is the economic–geographic nested matrix. Considering the influence of economic factors and geographical, the economic distance spatial weight matrix and geographical distance spatial weight matrix are nested to construct the matrix.

#### 3.1.4. Spatial Difference-in-Differences Model

The construction of smart cities is the primary driver of the digital economy [16]. The Ministry of Housing and Construction officially promulgated the smart cities list on 5 August 2013 (http://www.gov.cn/jrzg/2013-08/05/content_2461575.htm (accessed on 25 August 2022)) to promote the construction of smart city construction. To more robustly analyze the impact of the digital economy on PM_2.5_ emissions, this study constructed a dummy variable to investigate the SCP policy effect. The pilot city is 1, and the value of the non-pilot city is 0, which is used to analyze the impact of SCP policy on urban PM_2.5_ emissions sites. The spatial DID model has been set as follows.
(5)$$\begin{array}{cc} \ln pm_{it}= & \alpha_{0}+\beta \sum_{j=1}^{n}W_{ij}\ln pm_{it}+\gamma \ln(X_{it})+\lambda \sum_{j=1}^{n}W_{ij}\ln X_{it}\gamma_{1}+\gamma_{1}Smart\\ & +\lambda_{1}\sum_{j=1}^{n}Smart\gamma_{0}+\mu_{i}+\upsilon_{t}+\varepsilon_{it} \end{array}$$
where, *Smart* represents the dummy variable for whether SCP policy is implemented, and other symbols are identical as Equation (5).

A necessary prerequisite for the DID method is that the treatment and control groups satisfy the parallel trend hypothesis, i.e., there is either no significant difference in the digital economy and PM_2.5_ emissions between the treatment and control groups of cities before the SCP policy, or there is a relatively stable linkage trend. To address the characteristics of SCP policy implementation, this study adopted an event study approach to test the parallel trends.

The negative impact of the SCP policy on PM_2.5_ emissions may also come from some unobservable factors. To ensure the reliability of the estimation results, estimation bias due to the omission of explanatory variables needs to be eliminated. Xie et al. (2021) and Guo et al. (2022) employed a random sampling method for placebo tests. Specifically, we first divided all cities into the control group based on the actual SCP policy [16,41]. We then randomly selected the same number of cities in the sample as the treatment group and re-estimated benchmark estimates of 1000 times based on these placebo samples. The t-statistics and coefficient estimates of 1000 times followed essentially a normal distribution.

### 3.2. Variables Selected

The dependent variable is PM_2.5_ emissions per unit of production (*pm*). The Atmospheric Composition Analysis Group at Dalhousie University in Halifax combines global models, satellite observations, and air quality monitor data to develop estimates of on-the-ground PM_2.5_ levels (The resolution of the retrieval product is 0.01° × 0.01° spatial resolution. The data in the grid can be exported by ArcGIS software). The measured data are used in the empirical analysis of this paper.

The digital economy level was selected as the core independent variable. There is still no uniform standard for the digital economy level. Combining with existing studies on the digital economy [16,42,43,44], this study used the entropy method to measure the level of digital economy development for four indicators: digital infrastructure, digital industry development, digital innovation, and digital inclusive finance. Precisely, digital infrastructure is mainly measured by broadband internet infrastructure and mobile internet infrastructure by, respectively, the number of internet users and mobile phone users. The development level of the digital industry is mainly measured by the information industry basis and output value, which are the number of personnel in the information transmission, computer service, and software industry and the total amount of telecom business. Digital innovation level is mainly measured by spending on science and education. Digital financial inclusion is mainly measured from three aspects: coverage breadth index, use depth index, and digital measure, which are the coverage breadth index, use depth index, and digital measure index of digital financial. Finally, the entropy method is used to measure the comprehensive index of digital economy shown in Table 1. The development level of the city’s digital economic is divided into four stages by the natural break point method: lag phase, initial phase, propulsion phase, and leading phase.

Based on existing research and theoretical analysis, the economy development level (measured by per capita gross domestic product (GDP)) [7,45,46], urbanization level [47,48], urban size (measured by population) [49,50,51], investment in fixed assets [45,52,53], industrial upgrading [29,54,55], and green total factor productivity [56] are selected as control variables.

### 3.3. Data Sources

This study selected 283 prefecture-level cities in China from 2011 to 2019 as research samples. The data mainly comes from the Wind database, Guoyan web, China Urban Statistical Yearbook, Digital research center of Peking University, China Energy Statistical Yearbook, and China environmental Statistical Yearbook. Table 2 presents the descriptive statistics of a significant variable.

## 4. Spatial Evolution of the Digital Economy and PM_2.5_ Emissions

### 4.1. Agglomeration Characteristics of Digital Economy and PM_2.5_ Emissions

Based on the global Moran’s I index, this paper has analyzed the agglomeration characteristics of the digital economy development level and PM_2.5_ emissions in China, and further explored the spatial features. The results are shown in Table 2, which reports the global Moran’s I index for the digital economy [12,14] and PM_2.5_ emissions [40,57]. The Moran’s I indexes are all significantly greater than 0, indicating a positive spatial autocorrelation between the digital economy and PM_2.5_ emissions among Chinese cities. Specifically, the Moran’s I index of the digital economy is smaller than the Moran’s I index of PM_2.5_ emissions, indicating that the spatial autocorrelation of the digital economy is slightly weaker than the spatial autocorrelation of PM_2.5_ emissions.

To analyze the spatial relationship of the digital economy and PM2.5 emissions, Moran’s I indexes have been presented by different spatial distance thresholds of 100 km, 200 km, 300 km, 400 km, 500 km, and the entirety of these are in Table 3. The *p*-values of the Moran’s I index for the digital economy and PM_2.5_ emissions are zero, and the *z*-indexes are greater than 2.58, indicating significant spatial clustering of the digital economy and PM_2.5_ emissions in the city. Specifically, the Moran’s I indexes for the digital economy continues to decrease over time, and the Moran’s I indexes for PM_2.5_ emissions increase and decrease, suggesting that the correlation between different cities in the development of the digital economy has an impact on the spatial correlation of PM_2.5_ emissions. 

### 4.2. Spatial Distribution Characteristics of the Digital Economy

The spatial distribution characteristics of the digital economy were visualized by ArcGIS 10.6 software, as shown in Figure 1. From 2011 to 2019, the digital economy development level has shown a trend of improvement, as well as a form of aggregate development. The digital economy level was divided into four segments by the method of natural segment points. The development level of the digital economy below 0.047 is the I phase, between 0.048 and 0.095 is the II phase, between 0.096 and 0.142 is the III phase, and greater than 0.142 is the IV phase. The development of the digital economy in eastern coastal areas has shown an agglomeration effect, especially in the Pearl River Delta and Yangtze River Delta. In 2019, the development level of China’s digital economy significantly improved. The theme of the digital economy and economic development pattern shows the phenomenon of the core cities of digital economy spreading to surrounding cities. On one hand, the technology spillover effect is the reason. On the other hand, factor resource allocation optimization is the reason.

### 4.3. Spatial Dynamic Characteristics of the Digital Economy

To analyze the spatial dynamic characteristics, in this section, we used the Markov chain method to calculate the transition probability of the digital economy to analyze the spatial evolution characteristics of the digital economy between cities from 2011 to 2019. The digital economy development level is divided into four stages, including: I phase, II phase, III phase, and IV phase by the natural break point method. According to the results in Table 4, the diagonal values of II phase, III phase, and IV phase of digital economy development level are all greater than 0.75, indicating that there is a hierarchical solidification phenomenon in these three phases. The hierarchical solidification level of I phase is the weakest, with a probability of 0.384. The upper right of the main diagonal indicates the probability of the digital economy moving from a lower level to a higher level. Among them, the transfer possibility from the I phase to the II phase is the highest, which is 0.631. The second is from the II phase to III phase, with a probability of 0.222, followed by the transfer from propulsion phase to IV phase, with a probability of 0.173. In addition, the probability of cross-level transfer is minimal, which are less than 0.01, indicating that the development level of the urban digital economy in China has a steady and gradual trend, and multi-level leapfrog development is challenging to occur.

### 4.4. Spatial Transfer Characteristics of the Digital Economy

Due to spatial auto-correlation, this section analyzes the influence of geographical neighborhood relationships on the transition probability of digital economy development. The Markov transition matrix is used to analyze the transition probability of the digital economy development level under the influence of the community, and the results are shown in Table 5. In terms of the transfer probability of digital economy level in the lag phase’s neighborhood, the central diagonal values of IV phase, III phase, II phase, and I phase are 0.900, 0.600, 0.710, and 0.500 respectively, indicating that the probability of horizontal transfer of digital development in the IV phase is more negligible in the I phase community. From the perspective of transfer probability of digital economy level in the neighborhood of the II phase, the main diagonal values of the IV phase, III phase, II phase, and I phase are 0.750, 0.773, 0.750, and 0.500, respectively, indicating that the transfer probability of the II phase is strong. The digital economy level of I phase in the neighborhood of II phase has a possibility of 0.5 to move to a higher level. According to the perspective of the transfer probability of the digital economic development in the community of III phase, the central diagonal values of the IV phase, III, II phase, and I phase are 0.877, 0.750, 0.787, and 0.200, respectively, indicating that I phase in the neighborhood of the III phase has a significant influence on I phase. The probability of transfer to II phase is 0.800. In terms of the transfer probability of the digital economy level in the IV phase’s neighborhood, the central diagonal values of the IV phase, III phase, II phase, and I phase are 0.931, 0.799, 0.770, and 0.382, respectively, indicating that the probability of horizontal transfer of digital development in I phase is 0.618. According to the above analysis, the state transition of the digital economy has a specific spatial correlation, and, significantly, the I phase is most affected by the digital economy level of the surrounding cities. In addition, there are apparent differences in the influence of different levels of regions in the dynamic transfer. In particular, the IV phase can promote the joint development of the digital economy in surrounding cities.

### 4.5. Spatial Distribution Characteristics of PM_2.5_ Emissions

Figure 2 presented the spatial distribution characteristics of PM_2.5_ emissions for 283 cities in China for the three years 2011, 2015, and 2019, which were visualized by ArcGIS 10.6 software. The data characteristics of PM_2.5_ emissions was divided into five phases. Overall, PM2.5 is consistently higher in the central region and the Beijing-Tianjin-Hebei economic cluster than in other regions. Around 2011, the problem of haze in China threatened the health of the population. At one point, it became one of the most problematic issues in China. Overall, China’s PM2.5 emissions have decreased, indicating that China’s air quality has improved from 2011 to 2019.

## 5. The Impact of the Digital Economy on PM_2.5_ Emissions

### 5.1. Total Effect of the Digital Economy on PM_2.5_ Emissions

Based on the results of the above analysis, it was shown that digital economy and PM_2.5_ emissions have significant spatial correlation, and so ordinary least squares (OLS) analysis is challenging to estimate. To analyze the results reliably and robustly, a spatial Durbin model is used to analyze the impact of the digital economy on PM_2.5_ emissions in this paper. The estimation results are shown in Table 6. City fixed effects and time fixed effects are controlled in the model, as shown in City FE and Time FE in Table 6, respectively.

In Table 6, the neighborhood space matrix, the economic distance matrix, and the economic geographical nesting matrix were considered separately. The impact of the digital economy development on PM_2.5_ emissions was significantly negative under three types of weight, indicating that digital economy development significantly reduced the level of PM_2.5_ emissions in the region. This conclusion was consistent with [24,25,58]. Technological progress in the development of the digital economy has led to industrial upgrading [14,16,21], which in turn has driven energy restructuring and the gradual penetration of big data to promote the effectiveness of resource allocation [16,18,43]. At the same time, economic externalities were generated, leading to an increase in industrial productivity and urban energy utilization, ultimately contributing to the reduction of PM_2.5_ emissions.

The comparative analysis revealed differences in the coefficient statistics and significance of the digital economy under different spatial matrices. Even though the digital economy was not statistically significant under the neighborhood weights, it was significantly negative at alpha level of 0.05 under both the economic distance and the economic–geographic weight matrix. The marginal impact of the development of the digital economy on neighborhood PM_2.5_ emissions was more significant than the marginal impact on PM_2.5_ emissions in the region, which further illustrated the spatial relevance discussed in Section 4.

In control variables, there were also differences in the significance of the coefficients of several of the variables under different weighting matrices. Under the economic weights and the economic–geographic nested weight matrix, the level of economic development promotes the neighborhood PM_2.5_ emissions. Green total factor productivity, on the other hand, reduced PM_2.5_ emissions in the neighborhood.

### 5.2. Policy Effect of Smart Cities Pilot Policy

In 2013, the Chinese Ministry of Science and Technology and the National Standardisation Administration of China identified pilot cities for “smart city” technologies and standards. The distribution of pilot cities is shown in Figure 3. Through the application of new-generation information technologies such as Internet of Things (IOT) infrastructure, cloud computing infrastructure, and tools and processes such as wikis, social networks, Fab Lab, Living Lab, and integrated methods, smart cities achieve comprehensive and open creation through perception, broadband and ubiquitous interconnection, intelligent and integration. Along with the rise of network empires and the convergence of mobile technologies of innovation, the smart city in the knowledge society environment was the advanced form of informational city development after the digital city [16,59,60].

To evaluate the effect of SCP policy, this paper employed a spatial DID approach, and the estimated results are represented in Table 7. The classical DID model was first used for estimation and the results were shown in the first column. The estimated coefficient obtained for the SCP policy was significantly negative, indicating that smart city construction had reduced urban PM_2.5_ emissions to some extent. Further considering the spatial lag term of the SCP policy dummy variable, the coefficient of the *W×Smart* showed a significant positive effect. A possible reason was that SCP policy promoted low carbon, digital and sustainable high-quality development, causing a portion of high energy consumption and high emission industries to shift to neighboring cities [16]. Comparing the magnitude of the coefficients shows that the coefficients of *W×Smart* were the largest under the economic distance weighting matrix, indicating that this industrial shift prefers cities with similar levels of development.

### 5.3. Placebo Test

There may be some unobservable factors in the impact of SCP policy on PM_2.5_ emissions. This bias in estimation due to unobservable factors needs to be eliminated, and this study refers to Xie et al. (2021) and Guo et al. (2022) for a placebo trial using a random sampling method [16,41]. Specifically, all cities were divided into a control group based on the actual SCP policy. An identical number of cities in the sample were randomly selected as the treatment group. These samples were finally replicated an estimated 1000 times.

Kernel density plots of the t-statistic and estimated coefficients are reported separately in Figure 4. Based on results from Figure 4, the t-statistics and estimated coefficients of these 1000 regressions follow a normal distribution and the peak is around 0. Implying that the SCP policy has no significant effect on the randomly selected experimental group, further demonstrating the robustness of the study findings. 

### 5.4. Exclude the Influence of Other Policies

During the sample period of this paper, the Chinese government also introduced some other policies. In October 2014, China’s Ministry of Industry and Information Technology announced the list of pilot cities for Broadband China (BC) to speed up information transfer and improve the efficiency of social and economical operations. In December 2015, China’s Ministry of Industry and Information Technology (MIIT) announced the inclusion of 11 cities in the pilot list of regional industrial green transformation to promote the green and efficient development of industry. In 2017, the national development and reform commission (NDRC) issued the Notice on the Piloting of Low-carbon (LC) Provinces, Regions, and Cities to promote the development of low-carbon industries, build low-carbon cities, and advocate low-carbon living. These policies might also impact urban PM_2.5_ emissions.

To exclude the influence of these policies, this section controlled for them and presented the estimation results in Table 8. Columns (1)–(3) report the policy effect of BC, MITT, and LC pilot policies separately. The estimation results indicated that these three policies have no significant impact on PM_2.5_. Column (4) considered these three policies and the wise city pilot policy. The results show that the effects of digital economy development on urban PM_2.5_ concentration is still significantly negative. The coefficient of the smart city pilot policy is negative. However, the estimated coefficients of these three policies are not significant. 

## 6. Spatial Heterogeneity Analysis

Based on the uneven regional development of China, this section further analyzed the spatial variability of the impact of the digital economy on urban PM_2.5_ emissions. It was carried out in terms of both the geographical area (The division of the eastern, middle and western was mainly based on the criteria of the Nation Statistics Office. http://www.stats.gov.cn/xxgk/sjfb/zxfb2020/202207/t20220715_1886447.html (accessed on 25 August 2022)), and whether it was an urban agglomeration (China’s urban agglomerations include the Beijing-Tianjin-Hebei urban agglomeration, the Yangtze River midstream urban agglomeration, the Ha-Chang urban agglomeration, the Chengdu-Chongqing urban agglomeration, the Yangtze River Delta urban agglomeration, the Central Plains urban agglomeration, the Beibu Gulf urban agglomeration, the Guanzhong Plain urban agglomeration, the Hubao-Egyu urban agglomeration, the Lanzhou-West urban agglomeration and the Guangdong-Hong Kong-Macao Greater Bay Area). As the differences between regions include economic distance differences and geographical differences, they were calculated in the subsequent analysis based on an economic geographical nested matrix. The estimated results are shown in Table 9.

First, the impact of the digital economy on urban PM_2.5_ emissions is estimated for each of the three regions: eastern, middle, and western (The eastern region includes Beijing, Tianjin Hebei, Shanghai, Jiangsu, Zhejiang, Fujian, Guangdong, Shandong, Hainan, Liaoning, Jilin, and Heilongjiang. The middle region includes Shanxi, Anhui, Jiangxi, Henan, Hubei, and Hunan. The western region includes Guangxi, Chongqing, Sichuan, Yunnan, Tibet, Shaanxi, Gansu, Qinghai, Ningxia, and Xinjiang). In the eastern region, the development of the digital economy had significantly curbed PM_2.5_ emissions, as well as in neighboring areas. Each unit increase in the digital economy reduced PM_2.5_ emissions in the area by 7.25%, while reducing PM_2.5_ emissions in neighboring regions by 15.27%, all at an alpha level of 0.01. Since China’s reform and opening up, the eastern region had experienced rapid economic development and was at the leading edge of the country. The eastern region has more apparent advantages in infrastructure and digital industry development. In addition, the east area had gathered many innovative talents and capital, relying on various advantages to play an empowering role in the digital economy. The ultimate expression is green and emissions reduction [14,15]. In intermediate and west regions, the digital economy did not pass a significance test, probably because the level of development of the digital economy in the middle and west was still in its infancy. The increased resource consumption of the digital economy development and the effect of digital empowerment offset each other, ultimately resulting in a non-significant effect of the digital economy on PM_2.5_ emissions [14].

Second, from a city cluster perspective, the development of the digital economy had a mitigating effect on urban PM_2.5_ emissions within city clusters, and reduced neighboring cities through spillover effects. Especially, each unit increase in the digital economy of a city cluster reduced PM_2.5_ emissions by 4.02% in the region and by 6.17% in neighboring cities, both at alpha level of 0.05. It was mainly because the development of urban agglomerations showed strong synergies, and there were spillover effects between cities. For cities in non-urban clusters, the digital economy had no significant impact on PM_2.5_ emissions, but reduced PM_2.5_ emissions in neighboring cities.

In summary, the spatial heterogeneity analysis suggested that the environmental dividends of the digital economy in eastern and urban agglomerations are more fully realized in the form of reduced PM_2.5_ emissions in the region and neighboring areas. The cities in the central and west and non-urban clusters did not pass the significance test. The possible reason for this is that the level of digital economy development is still early in the central and west and non-urban cluster cities. The increase in resource consumption caused by the initial development of the digital economy is offset by the boost generated by digital empowerment.

## 7. Conclusions

To investigate the spatial effect of digital economy on PM_2.5_ emissions in China, this paper establishes a spatial Durbin model by using prefecture-level cities from 2011 to 2019. We further discuss the policy effect and spillover effect of smart city pilot policy. In addition, we analyze the heterogeneous results between the digital economy and PM_2.5_ emissions in two aspects. Accordingly, we highlight the following conclusions:

First, urban digital economy development is significantly negatively correlated with PM_2.5_ emissions. Specifically, a 1% increase in the digital economy index will mitigate PM_2.5_ emissions by an average of 2.44%. The spatial spillover effect results indicate that the adverse spillover effects of the digital economy on neighboring PM_2.5_ emissions are more substantial. A 1% increase in the digital economy index will mitigate neighboring PM_2.5_ emissions by an average of 4.99%. Possible reasons for this are that the development of the digital economy promotes technological efficiency and reduces PM_2.5_ emissions in the region and neighboring regions through technological spillover [61]. That is to say, digital economy development significantly facilitates the process of green economic growth.

Second, considering the policy effect of the smart city pilot policy, this paper uses classical DID and spatial DID models to explore the impact of smart city construction on PM_2.5_ emissions. The empirical results persisted that SCP policy significantly reduces PM_2.5_ emissions of pilot cities than non-pilot cities, and the spatial DID results show that SCP policy significantly increases PM_2.5_ emissions of the neighbor cities of pilot cities. One possible reason is that the construction of smart cities has led to the transfer of highly polluting and emitting industries from the pilot cities to neighboring cities, thereby increasing PM_2.5_ emissions from neighboring cities. In other words, smart city construction reduces PM_2.5_ emissions in the pilot cities but increases PM_2.5_ emissions in neighboring cities. To achieve green and sustainable development, the scope of smart cities should be promoted, thus improving environmental quality.

Third, because of the uneven development of China’s regional economies and the differences in the development of urban agglomerations, the differences are analyzed in these two aspects. In the eastern region, a 1% increase in the digital economy index will mitigate PM_2.5_ emissions by an average of 7.25% and 15.27% in region and neighboring cities, respectively. For cities in urban agglomerations, a 1% increase in the digital economy index will mitigate PM_2.5_ emissions by an average of 4.02% and 6.17 in region and neighboring cities, respectively. In other words, the digital economy spillover effect from eastern and urban agglomeration should be given full play to drive the development of the digital economy in middle and western regions and non-urban agglomerations, thereby promoting green and sustainable economic development.

According to the above conclusions, we put forward the following insights. First, the process of smart city construction, the development of the digital economy is conducive to reducing PM_2.5_ emissions, thus promoting green economic development. Second, the spatial spillover effects of the digital economy from eastern cities and urban agglomerations are still insufficient to affect central and western cities and non-urban agglomerations, illustrating the clear imbalance that characterizes the development of China’s digital economy. In a word, the development of digital economy in urban regions still has great potential for the digital economy in China.

## Figures and Tables

**Figure 1 ijerph-19-14456-f001:**
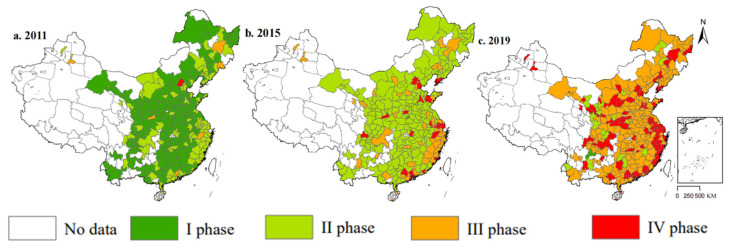
Spatial-temporal evolution pattern of China’s digital economy. *Note*: Drawn by authors using ArcGIS 10.6.

**Figure 2 ijerph-19-14456-f002:**
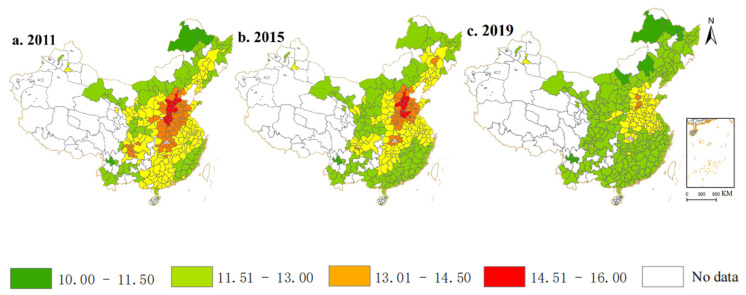
Spatial-temporal evolution pattern of China’s PM_2.5_ emissions. *Note*: Drawn by authors using ArcGIS 10.6. (**a**–**c**) are the serial number of the pictures.

**Figure 3 ijerph-19-14456-f003:**
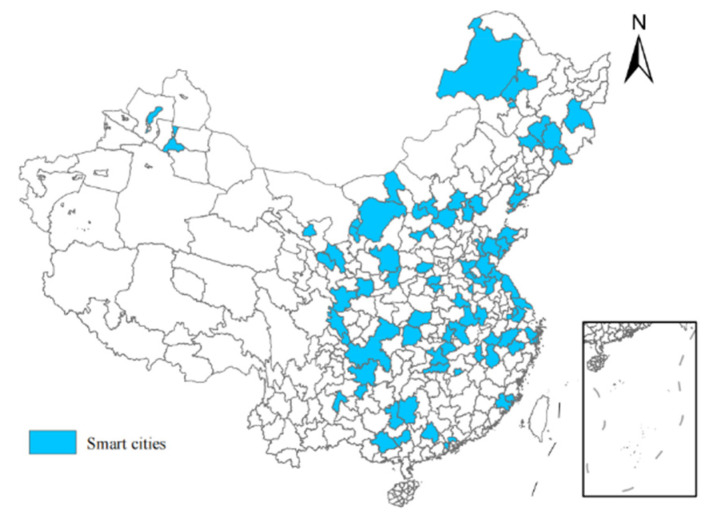
Smart cities pilot.

**Figure 4 ijerph-19-14456-f004:**
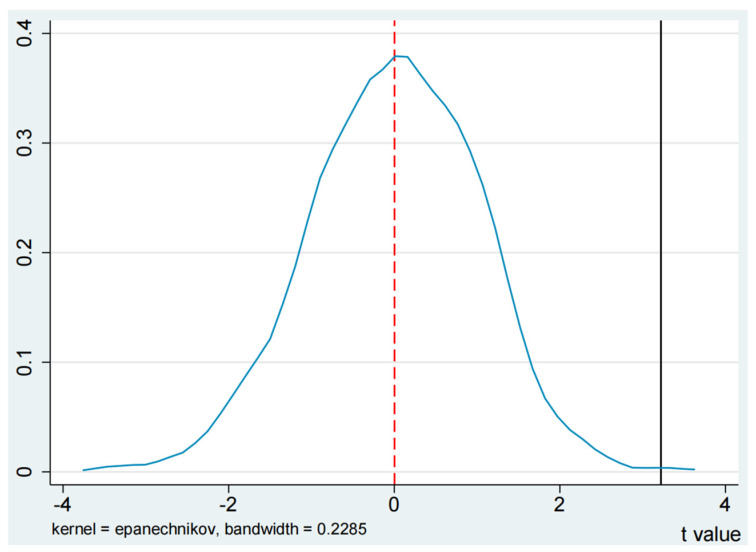
Placebo test. *Note*: The curves indicate the distribution of t-values and estimated coefficients for 1000 replicate tests, respectively. The red dashed line indicates that the results of 1000 repeated experiments basically obey the state distribution.

**Table 1 ijerph-19-14456-t001:** Comprehensive index system of digital economy.

Primary Index	Secondary Index	Tertiary Index	Indicator Description	Index Attribute
Digital economy	Digital infrastructure	Broadband internet	Number of internet users (per 10,000 people)	+
		Mobile internet	Numbers mobile phone users (per 10,000 people)	+
	Digital industry development	Information industry basis	Number of personnel in the information transmission, computer service and software industry	+
		Information industry output value	Total amount of telecom business (CNY 10,000)	+
	Digital innovation	Digital innovation factor support	Spending on science and education (%)	+
	Digital financial inclusion	Coverage breath index	Coverage breadth index of digital financial	+
		Use depth index	Use depth index of digital financial	+
		Digital measure	Digital measure index of digital financial	+

**Table 2 ijerph-19-14456-t002:** Descriptive statistics.

Explanation of the variables	Data Sources	Symbol	Obs	Mean	Sd.	Min	Max	Unit
*PM_2.5_ emissions*	*Atmospheric Composition Analysis Unit, Dalhousie University*	*ln* pm	2547	43.6140	15.1208	13.8656	108.5263	Microgramme/stere
*Digital economy*	*Digital research center of Peking University*	*ln* Digital	2547	0.0949	0.0552	0.0102	0.8199	/
*Per GDP*	*China Urban Statistical Yearbook*	*ln* AGDP	2547	34,464.23	29,980.83	99	467,749	CNY/year
*Urbanization level*	*China Urban Statistical Yearbook*	*ln* Urb	2547	0.5496	0.1474	0.1815	1	/
*Population size*	*China Urban Statistical Yearbook*	*ln* Popu	2547	447.4903	319.9367	19.5	3404.01	/
*Fixed asset investment ratio*	*China Urban Statistical Yearbook*	*ln* Invest	2547	0.0033	0.0035	0.0001	0.0406	/
*Industrial upgrading*	*China Urban Statistical Yearbook*	*ln* Up	2547	0.3585	4.1002	0.0001	206.934	/
*Green total factor productivity*	*Guoyan web and Wind database*	*ln GTFP*	2547	1.5371	0.7476	0.1044	2.9482	/

*Note:* In the follow regression, logarithms of all variables are taken.

**Table 3 ijerph-19-14456-t003:** Moran’s I index of the digital economy and PM_2.5_ emissions at different spatial thresholds.

		2011	2012	2013	2014	2015	2016	2017	2018	2019
*ln* Digital	*entirety*	0.220 *** (5.966)	0.199 *** (5.412)	0.179 *** (4.895)	0.155 *** (4.251)	0.151 *** (4.131)	0.151 *** (4.137)	0.150 *** (4.114)	0.149 *** (4.093)	0.138 *** (3.795)
	*D100*	0.290 *** (6.656)	0.234 *** (5.232)	0.266 *** (5.956)	0.242 *** (5.428)	0.213 *** (4.784)	0.218 *** (4.885)	0.227 *** (5.084)	0.231 *** (5.175)	0.220 *** (4.928)
	*D200*	0.178 *** (8.239)	0.138 *** (6.437)	0.155 *** (7.187)	0.119 *** (5.556)	0.117 *** (5.489)	0.111 *** (5.489)	0.123 *** (5.198)	0.106 *** (4.974)	0.100 *** (4.713)
	*D300*	0.105 *** (7.291)	0.084 *** (5.931)	0.086 *** (6.076)	0.057 *** (4.101)	0.059 *** (4.216)	0.051 *** (3.665)	0.052 *** (3.734)	0.051 *** (3.709)	0.040 ** (2.939)
	*D400*	0.070 *** (6.625)	0.054 *** (6.625)	0.056 *** (5.332)	0.031 *** (3.316)	0.035 *** (3.446)	0.034 *** (3.339)	0.038 *** (3.770)	0.028 ** (2.823)	0.026 ** (2.682)
	*D500*	0.056 *** (6.692)	0.043 *** (5.260)	0.047 *** (5.669)	0.021 ** (2.788)	0.020 ** (2.671)	0.028 *** (3.540)	0.030 *** (3.763)	0.022 ** (2.823)	0.022 ** (2.855)
*ln* pm	*entirety*	0.273 *** (7.394)	0.279 *** (7.550)	0.261 *** (7.091)	0.266 *** (7.205)	0.290 *** (7.855)	0.286 *** (7.7500)	0.276 *** (7.468)	0.267 *** (7.248)	0.269 *** (7.294)
	*D100*	0.202 *** (4.549)	0.219 *** (4.908)	0.205 *** (4.608)	0.193 *** (4.346)	0.224 *** (5.025)	0.226 *** (5.073)	0.204 *** (4.592)	0.198 *** (4.442)	0.211 *** (4.737)
	*D200*	0.123 *** (5.749)	0.130 *** (6.055)	0.116 *** (5.431)	0.093 *** (4.375)	0.129 *** (6.024)	0.131 *** (6.106)	0.112 *** (5.238)	0.108 *** (5.044)	0.121 *** (5.660)
	*D300*	0.109 *** (7.603)	0.107 *** (7.461)	0.108 *** (7.507)	0.080 *** (5.662)	0.117 *** (8.143)	0.117 *** (8.117)	0.101 *** (7.030)	0.105 *** (7.321)	0.108 *** (7.506)
	*D400*	0.103 *** (9.592)	0.103 *** (9.595)	0.099 *** (9.202)	0.079 *** (7.410)	0.109 *** (10.092)	0.111 *** (10.329)	0.101 *** (9.375)	0.107 *** (9.906)	0.104 *** (9.656)
	*D500*	0.084 *** (9.844)	0.084 *** (9.919)	0.076 *** (8.893)	0.065 *** (7.711)	0.087 *** (10.240)	0.088 *** (10.335)	0.081 *** (9.582)	0.085 *** (10.017)	0.081 *** (9.481)

*Note:* Z-value in parentheses. ** *p* < 0.05, *** *p* < 0.01. D100, D200, D300, D400, and D500 indicate 100 km, 200 km, 300 km, 400 km, and 500 km space thresholds, respectively.

**Table 4 ijerph-19-14456-t004:** Markov transition probability of digital economy from 2011 to 2019.

	Obs.	I Phase	II Phase	III Phase	IV Phase
I phase	297	0.384	0.613	0.003	0
II phase	1208	0.003	0.769	0.222	0.006
III phase	556	0	0.040	0.788	0.173
IV phase	203	0	0.010	0.069	0.921

**Table 5 ijerph-19-14456-t005:** Spatial Markov transition probability of the digital economic level.

		Obs.	I Phase	I Phase	III Phase	IV Phase
I phase	I phase	6	0.500	0.500	0.000	0.000
	II phase	31	0.000	0.710	0.290	0.000
	III phase	25	0.000	0.080	0.600	0.280
	IV phase	10	0.000	0.000	0.100	0.900
II phase	I phase	14	0.500	0.500	0.000	0.000
	II phase	40	0.000	0.750	0.250	0.000
	III phase	22	0.000	0.000	0.773	0.227
	IV phase	4	0.000	0.000	0.250	0.750
III phase	I phase	10	0.200	0.800	0.000	0.000
	II phase	75	0.000	0.787	0.213	0.000
	III phase	36	0.000	0.028	0.750	0.222
	IV phase	15	0.000	0.000	0.133	0.877
IV phase	I phase	267	0.382	0.614	0.004	0.000
	II phase	1062	0.004	0.770	0.219	0.007
	III phase	473	0.000	0.040	0.799	0.161
	IV phase	174	0.000	0.011	0.057	0.931

**Table 6 ijerph-19-14456-t006:** Impact of the digital economy on PM _2.5_ emissions.

Variables	Neighborhood Weight		Economic Weight		Economic−Geographical Nested	
	*ln* pm	W_1_×*ln* pm	*ln* pm	W_2_×*ln* pm	*ln* pm	W_3_×*ln* pm
*ln* Digital	−0.0416 ** (0.0175)	0.0109 (0.0134)	−0.0322 ** (0.0130)	−0.1118 *** (0.0272)	−0.0244 ** (0.0117)	−0.0499 ** (0.0218)
*ln* AGDP	0.0030 (0.0042)	0.0086 (0.0061)	−0.0061 (0.0069)	0.0745 ** (0.0290)	−0.0047 (0.0052)	0.0394 ** (0.0146)
*ln* Urb	−0.0306 ** (0.0156)	−0.0110 (0.0269)	−0.0492 (0.0323)	−0.2178 ** (0.0832)	−0.0411 (0.0290)	−0.1106 (0.0806)
*ln* Popu	−0.0207 (0.0233)	−0.0950 (0.0758)	−0.0039 (0.0338)	0.1952 ** (0.0906)	0.0001 (0.0298)	0.0409 (0.0890)
*ln* Invest	−0.0047 ** (0.0017)	−0.0005 (0.0026)	−0.0066 * (0.0039)	0.0207 ** (0.0092)	−0.0063 * (0.0034)	0.0078 (0.0076)
*ln* Up	0.0059 (0.0039)	−0.0002 (0.0046)	0.0064 (0.0050)	0.0204 ** (0.0097)	0.0053 (0.0043)	0.0165 ** (0.0076)
*ln* GTFP	−0.0010 (0.0023)	−0.0051 * (0.0027)	0.0013 (0.0050)	−0.0258 *** (0.0061)	0.0023 (0.0043)	−0.0151 ** (0.0050)
rho	0.8811 *** (0.0178)		0.5512 *** (0.0244)		0.7352 *** (0.0206)	
City FE	Yes		Yes		Yes	
Year FE	Yes		Yes		Yes	
Obs.	2547		2547		2547	
R^2^	0.6608		0.7031		0.7051	
likelihood	2929.6137		1756.5161		1993.1250	

*Note:* Standard errors in parentheses. * *p* < 0.1, ** *p* < 0.05, *** *p* < 0.01.

**Table 7 ijerph-19-14456-t007:** Policy effect of SCP policy.

Variables	Traditional DID	Spatial DID		
	(1)	(2)	(3)	(4)
*Smart*	−0.0431 *** (0.0153)	0.0012 (0.0052)	0.0108 (0.0141)	0.0028 (0.0124)
*W* *×Smart*		0.0396 *** (0.0117)	0.0700 ** (0.0293)	0.0501 * (0.0264)
*ln* Digital	−0.2849 *** (0.0127)	−0.0411 *** (0.0106)	−0.0988 *** (0.0112)	−0.0588 *** (0.0096)
*ln* AGDP	0.01559 * (0.0087)	0.0027 (0.0035)	−0.0080 (0.0058)	−0.0050 (0.0046)
*ln* Urb	−0.2146 *** (0.0440)	−0.0375 ** (0.0132)	−0.0888 ** (0.0323)	−0.0662 ** (0.0284)
*ln* Popu	−0.1104 (0.0779)	−0.0249 (0.0220)	−0.0111 (0.0363)	0.0001 (0.0296)
*ln* Invest	−0.0118 ** (0.0045)	−0.0049 ** (0.0017)	−0.0104 ** (0.0039)	−0.0077 ** (0.0033)
*ln* Up	0.0139 ** (0.0066)	0.0057 (0.0036)	0.0065 (0.0048)	0.0051 (0.0041)
*ln* GTFP	−0.0373 *** (0.0027)	−0.0027 (0.0013)	−0.0082 *** (0.0024)	−0.0040 * (0.0022)
Fixed effect	Yes	Yes	Yes	Yes
Obs.	2547	2547	2547	2547
R^2^	0.5890	0.6974	0.6741	0.6846

*Note:* Standard errors in parentheses. * *p* < 0.1, ** *p* < 0.05, *** *p* < 0.01. W indicates spatial weight matrix.

**Table 8 ijerph-19-14456-t008:** Excluding the other policies effects.

Variables	(1)		(2)		(3)		(4)	
	*ln* pm	W×*ln* pm	*ln* pm	W×*ln* pm	*ln* pm	W×*ln* pm	*ln* pm	W×*ln* pm
*ln* Digital	−0.0403 ** (0.0176)	0.0089 (0.0132)	−0.0405 ** (0.0175)	0.0102 (0.0134)	−0.0391 ** (0.0172)	0.0147 (0.0137)	−0.0377 ** (0.0173)	0.0107 (0.0136)
*smart*							−0.0095 * (0.0050)	0.0136 (0.0094)
BC	−0.0041 (0.0064)	0.0146 (0.0101)					−0.0029 (0.0063)	−0.0103 (0.0122)
RIGT			0.0046 (0.0120)	0.0043 (0.0205)			0.0035 (0.0116)	0.0027 (0.0200)
LC					−0.0114 (0.0087)	−0.0157 * (0.0083)	−0.0097 (0.0087)	−0.0173 ** (0.0083)
*ln* AGDP	0.0028 (0.0042)	0.0088 (0.0062)	0.0029 (0.0042)	0.0086 (0.0062)	0.0025 (0.0040)	0.0030 (0.0060)	0.0025 (0.0040)	0.0038 (0.0061)
*ln* Urb	−0.0284 ** (0.0144)	−0.0142 (0.0255)	−0.0253 * (0.0147)	−0.0175 (0.0259)	−0.0292 * (0.0152)	−0.0212 (0.0259)	−0.0356 ** (0.00147)	−0.0157 (0.0259)
*ln* Popu	−0.0209 (0.0236)	−0.0970 (0.0780)	−0.0196 (0.0233)	−0.0935 (0.0746)	−0.0173 (0.0235)	−0.0921 (0.0737)	−0.0160 (0.0226)	−0.0927 (0.0758)
*ln* Invest	−0.0047 ** (0.0017)	−0.0008 (0.0027)	−0.0047 ** (0.0017)	−0.0009 (0.0026)	−0.0047 ** (0.0017)	−0.0006 (0.0027)	−0.0050 *** (0.0016)	−0.0003 (0.0027)
*ln* Up	0.0058 (0.0039)	−0.0004 (0.0046)	0.0059 (0.0039)	−0.0005 (0.0045)	0.0060 (0.0037)	0.0005 (0.0046)	0.0057 (0.0038)	−0.0002 (0.0045)
*ln* GTFP	−0.0012 (0.0023)	−0.0053 * (0.0028)	−0.0011 (0.0023)	−0.0049 * (0.0027)	−0.0012 (0.0023)	−0.0050 * (0.0027)	−0.0012 (0.0023)	−0.0059 ** (0.0028)
rho	0.8831 *** (0.0176)		0.8832 *** (0.01760)		0.8717 *** (0.0196)		0.8721 *** (0.0197)	
Controls	Yes		Yes		Yes		Yes	
City FE	Yes		Yes		Yes		Yes	
Year FE	Yes		Yes		Yes		Yes	
Obs.	2547		2547		2547		2547	
R^2^	0.6661		0.6626		0.7122		0.7227	
likelihood	2956.1962		2954.8262		2968.6545		2975.3857	

*Note:* Standard errors in parentheses. * *p* < 0.1, ** *p* < 0.05, *** *p* < 0.01. BC presents Broadband China pilot policy. RIGT presents regional industrial green transfer pilot policy. LC presents low-carbon pilot policy.

**Table 9 ijerph-19-14456-t009:** Regression results for the spatial heterogeneity test.

Variables	East		Central		West		Urban Agglomerations		Non−Urban Cluster	
	*ln* pm	W×*ln* pm	*ln* pm	W×*ln* pm	*ln* pm	W×*ln* pm	*ln* pm	W×*ln* pm	*ln* pm	W×*ln* pm
*ln* Digital	−0.0725 *** (0.0207)	−0.1527 *** (0.0474)	0.0176 (0.0351)	−0.0661 (0.0866)	−0.0456 (0.0281)	−0.0318 (0.0422)	−0.0402 ** (0.0187)	−0.0617 ** (0.0310)	−0.0213 (0.0153)	−0.0651 ** (0.0274)
*ln* AGDP	0.1528 ** (0.0711)	−0.0865 (0.0674)	−0.0286 (0.0375)	0.0389 (0.0487)	0.0155 (0.0177)	0.0214 (0.0166)	−0.0098 (0.0066)	0.0547 ** (0.0180)	0.0016 (0.0129)	0.736 *** (0.0192)
*ln* Urb	0.1196 * (0.0648)	0.0321 (0.1199)	−0.1918 (0.1543)	−0.2963 (0.2724)	−0.0534 (0.0554)	−0.5207 ** (0.1772)	−0.0172 (0.0342)	−0.2259 ** (0.0863)	−0.0435 (0.0384)	−0.0658 (0.0942)
*ln* Popu	0.730 *** (0.1861)	−0.1387 (0.2662)	0.4217 *** (0.1167)	−0.1357 (0.1506)	1.0427 (0.0411)	−0.3568 *** (0.0879)	0.0115 (0.0278)	0.0877 (0.05580)	0.0177 (0.0655)	−0.3199 * (0.1891)
*ln* Invest	0.0073 (0.0082)	−0.0007 (0.0138)	−0.0137 * (0.0080)	0.0177 (0.0142)	0.0024 (0.0071)	0.0251 (0.0155)	−0.0098 ** (0.0049)	0.0172 * (0.0102)	−0.0023 (0.0041)	−0.0043 (0.0085)
*ln* Up	0.0312 ** (0.0147)	0.0253 (0.0176)	−0.0198 (0.0138)	0.0317 (0.0328)	0.0145 (0.0114)	−0.0632 ** (0.0262)	−0.0026 (0.0082)	0.0184 (0.0122)	0.0106 ** (0.0039)	0.0022 (0.0126)
*ln* GTFP	−0.0576 ** (0.0274)	0.0190 (0.0238)	0.0166 (0.0129)	−0.0355 ** (0.0139)	−0.0515 *** (0.0100)	0.0378 *** (0.0110)	0.0045 (0.0063)	−0.0244 *** (0.0075)	−0.0056 *** (0.0055)	−0.0079 (0.0064)
rho	0.5854 *** (0.0532)		0.6401 *** (0.0644)		0.5305 *** (0.0483)		0.6719 *** (0.0320)		0.7225 *** (0.0284)	
City FE	Yes		Yes		Yes		Yes		Yes	
Year FE	Yes		Yes		Yes		Yes		Yes	
Obs.	909		891		747		1134		1413	
R^2^	0.5507		0.2953		0.2480		0.7672		0.6474	
likelihood	517.8134		364.7632		463.4475		899.4301		1138.3396	

*Note:* Standard errors in parentheses. * *p* < 0.1, ** *p* < 0.05, *** *p* < 0.01.

## Data Availability

It can be obtained from the author on request.

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
