# Peer review of "Spatial Effect of Digital Economy on Particulate Matter 2.5 in the Process of Smart Cities: Evidence from Prefecture-Level Cities in China"

_ijerph, 2022, doi:10.3390/ijerph192114456_

Round 1
Reviewer 2 Report
This paper studies the relationship between the digital economy development and fine particulate matter (PM2.5) emissions in 283 prefecture-level cities in China from 9 years of annual air quality and socioeconomic data. The Spatial Durbin model was used to explore the spatial spillover effect of the digital economy on PM2.5, and a spatial difference-in-differences (SDID) model was used to analyze the impact of the smart city pilot (SCP) policy on PM2.5. The research topic is highly interesting and the content of the manuscript is logically written. However, several issues need to be addressed before considering acceptance for publication. Please see my detailed comments below:
1. Consider including data during the COVID-19 pandemic period in the analysis. The authors emphasize the impact of COVID-19 to boost the development of the digital economy and its possible impact on PM2.5 in China. This statement seems the driver of this study. However, only 2011-2019 years annual data were used in the Spatial Durbin model and SDID model to link the connection between those two. It would be nice to include newer available data from 2020-2022 into the analysis or even nicer to compare the possible distinction pattern before COVID-19 and after COVID-19.
2. Line 226, equation 4. The subscript t in the equation means a year with the range [1 9] in your case?
3. Line 268-271. Need the citation of the Dalhousie PM2.5 emission product (now Dr. Martin moves to Washington University in St. Louis). What’s the resolution of the retrieval product used? Need clarification on how to map the gridded information into a signal PM2.5 emission value on the prefectural-level city.
4. Table 2. Even though equations 4 and 5 use the natural logarithm format of the data, it would be nice to present the input data as comment format, also need to include the unit of the input data such as emission in Ttons/year, GDP in dollars, etc.
5. Table 3. This table is busy without clear identification of the variables (e.g. Dxxx means different spatial distance thresholds). Suggest condensing or changing another format to present.
6. Line 374, Table 3? Typo?
7. Figure 2. Is it true that the PM25 emission hotspots in HulunBuir of Inner Mongolia and Jiuquan in Gansu province as shown in Figure 2? This seems not to align with socioeconomic development and other high-resolution emission data such as MEIC (Li et al., 2017). Again, see my comment #4, is it misleading to use the logarithm value to plot the emission data.
8. Table 6. The title needs to revise to reflect the most accurate information the table present. This comment applies to all Figures and Tables of this manuscript, please keep the caption and variable concise but clearly explained so that the reader can understand the content as the standalone version even without looking in depth at the corresponding content.
9. Line 522. Define the “eastern, middle and western” regions on urban PM2.5 emissions.
10. Table 9, what are the “city FE” and “Year FE” here?
11. Line 603-604. I don’t follow closely the publication requirement of the IJERPH, but normally we present the key supplementary and supporting materials along with the main content manuscript.
Reference:
Li, M., Liu, H., Geng, G., Hong, C., Liu, F., Song, Y., Tong, D., Zheng, B., Cui, H., Man, H. and Zhang, Q., 2017. Anthropogenic emission inventories in China: a review. National Science Review, 4(6), pp.834-866.
